# Pine Resin Derivatives as Sustainable Additives to Improve the Mechanical and Thermal Properties of Injected Moulded Thermoplastic Starch

**Miguel Aldas** [1,2,*], **Cristina Pavon** [2], **Juan López-Martínez** [2] and **Marina Patricia Arrieta** [3,*]

1   Departamento de Ciencia de Alimentos y Biotecnología, Facultad de Ingeniería Química y Agroindustria, Escuela Politécnica Nacional (EPN), Quito 170517, Ecuador
2   Instituto de Tecnología de Materiales, Universitat Politècnica de València (UPV), 03801 Alcoy-Alicante, Spain; crisppavonv@gmail.com (C.P.); jlopezm@mcm.upv.es (J.L.-M.)
3   Facultad de Óptica y Optometría, Universidad Complutense de Madrid (UCM), Arcos de Jalón 118, 28037 Madrid, Spain
*   Correspondence: miguel.aldas@epn.edu.ec (M.A.); marrie06@ucm.es (M.P.A.);
    Tel.: +593-999-736-444 (M.A.); +34-913-946-885 (M.P.A.)

**Abstract:** Fully bio-based materials based on thermoplastic starch (TPS) were developed starting from corn starch plasticized with glycerol. The obtained TPS was further blended with five pine resin derivatives: gum rosin (GR), disproportionated gum rosin (dehydroabietic acid, RD), maleic anhydride modified gum rosin (CM), pentaerythritol ester of gum rosin (LF), and glycerol ester of gum rosin (UG). The TPS–resin blend formulations were processed by melt extrusion and further by injection moulding to simulate the industrial conditions. The obtained materials were characterized in terms of mechanical, thermal and structural properties. The results showed that all gum rosin-based additives were able to improve the thermal stability of TPS, increasing the degradation onset temperature. The carbonyl groups of gum rosin derivatives were able to interact with the hydroxyl groups of starch and glycerol by means of hydrogen bond interactions producing a significant increase of the glass transition temperature with a consequent stiffening effect, which in turn improve the overall mechanical performance of the TPS-resin injected moulded blends. The developed TPS–resin blends are of interest for rigid packaging applications.

**Keywords:** bioplastic; corn starch; glycerol; thermoplastic starch; gum rosin; injection-moulding

## 1. Introduction

Polymers have become essential materials in our lives mainly due to their unique properties featuring lightness and durability and, as a consequence, their consumption has increased during the last decades [1]. However, the mass production of plastics and the limitation of non-renewable sources have led to problems with their final disposal and in end-of-life options [1,2]. This has promoted the search for alternatives to the use of fossil-based polymers with materials that present better environmental performance, particularly for short term applications (i.e., food packaging, disposable cutlery, agricultural applications) [2,3]. In recent years, biopolymers have risen as an alternative to traditional plastics, mainly for short term applications. Many research and industrial efforts have been focused on the development of sustainable polymers, mostly for single use disposal applications such as bottles, cold drink cups, thermoformed trays and container lids, blister packages, overwrap as well as flexible films, which are currently commercialized [2–4]. However, many commercially available biopolymers possess a low glass transition temperature, above which the polymeric matrix loses its

rigidity leading to plastic deformation, which make them unsuitable for hot beverages and/or hot food applications.

Biopolymers are materials whose synthesis involves micro-organisms and that have a reduced lifespan after final disposition [5,6]. Biopolymers can be produced naturally or obtained from biomass. They can be produced from: (i) wood derivatives such as cellulose [7,8], lignin [9] or terpenes [10], (ii) lipids as fatty acids [11] and (iii) polysaccharides as sugar [12], chitin, chitosan [12] or starch [3,13]. Polysaccharides account for 75% of the world's annual biomass production, with 170 trillion metric tons. Starch is the second most abundant and available polysaccharide in nature [14]. It is composed of two D-glucose polysaccharides: amylose, a linear polymer and amylopectin, a branched polymer [3]. Starch can be obtained from various sources including cassava, wheat, rice, potato, pea, and corn. Corn is the largest source of starch in the world [3,14]. Starch is widely used in the production of biopolymers given its abundance and low cost [3,15]. Starch is used as a main polymeric matrix as well as minor component or filler in polymer blends [16]. Moreover, some bioplastics in the market are produced from starch, such as poly(lactic acid) (PLA) [2,4]. Starch in its native state does not exhibit a thermoplastic behaviour. Hence, to use starch in the plastic industry it is necessary to break down the structure of granular starch [17]. The molecular order of the starch granules is destroyed by converting its molecular structure to thermoplastic through the gelatinization process, changing its crystalline structure to an amorphous structure [14]. To produce plasticization in the starch structure these features are required: (1) temperatures greater than 70 °C, (2) presence of plasticizers (water or polyols) and, (3) shear stress [17,18]. These conditions can be achieved in the extrusion process, which will allow an amorphous paste, known as thermoplastic starch (TPS) to be obtained, where most of the inter-macromolecular hydrogen bonds of starch are destroyed [3]. TPS can melt and flow allowing it to be processed by conventional moulding and extrusion processing techniques used for most thermoplastic polymers [18]. Glycerol is the most widely used plasticizer for the production of TPS [3]. From an environmental point of view, glycerol is a by-product of biodiesel production and the revalorization of glycerol increases its added value from a low-grade by-product to a useful biopolymer plasticizer [19]. TPS has gained industrial attention, compared with other biopolymers, due to its high availability and low cost, being one of the cheapest biopolymers in the market [5,15,16]. In fact, the next generation of packaging materials should fulfil the requirements to support the transition to a circular economy, which comprise a ban on single-use traditional plastics by the end of 2020 [20,21]. Moreover, from an environmental point of view, TPS is not only bio-based but it is also biodegradable, allowing the loop of circular economy to be closed. However, TPS industrial application is limited by its fragility, low water resistance, and the re-crystallization and retrogradation that its structure is subject to, which leads to undesirable changes in the thermomechanical properties of the material [5,17]. In this context, several strategies have been focused on improving biopolymer performance for extending its industrial applications in the packaging field, such as blending, copolymerization and/or the addition of either micro- or nano-fillers for the development of composites and nanocomposites [2,22]. Among all strategies, physical blending seems to be one of the most convenient routes to create new bio-based polymeric formulations, with the required combination of properties, by simply mixing two or more bio-based polymeric matrices in the melt state with no chemical reactions taking place [8,23]. Thus, melt-blending strategies have many advantages since they offer the opportunity to improve the final performance of biopolymers through relatively simple and readily available processing methods that are cost-effective for the industrial sector with respect to the development of new starting monomers and/or polymerization techniques [8,23]. Therefore, to improve TPS performance and extend its industrial applications, the blending approach has been widely studied. In fact, TPS have been blended with either fossil-based polymers (i.e., polypropylene (PP) [24], polystyrene (PS) [25], polyethylene (PE) [26] and ethylene-vinyl acetate (EVA) [27]) as well as with other biopolymers (i.e., PLA [28], and polyhydroxybutyrate (PHB) [29]) to improve its performance and increase its range of applications.

Gum rosin (GR) is a natural alternative to fossil-based polymers obtained from heating and evaporation of pine resin [30–32]. It possesses many advantages for the plastic industry since it

is abundant in the nature, has a relatively low cost, and is easily converted into high performance macromolecular materials [30,33]. Untreated rosin is rigid and brittle and has a thermoplastic behaviour. It is insoluble in water and soluble in acids and most organic solvents (i.e., glacial acetic acid, chloroform, ether and alcohol) [30]. GR is composed of 10–20% of neutral compounds and 80–90% of resin acids [32]. Resin acids structure is composed by a tricyclic skeleton with two double bonds and a carboxylic functional group. This structure provides GR the possibility to be chemically modified in order to obtain a wide range of derivatives, including polymerizable structures for both linear and crosslinked materials [10].

Chemical derivatives of GR can be obtained from oxidation, hydrogenation, dehydrogenation, isomerization, Diels-Alder couplings, esterification, saponification reactions with formaldehyde and phenol [30,32,34–36]. These GR derivatives can be used for the synthesis of homopolymers and copolymers with traditional monomers in order to modulate the properties of the resulting material [35]. In our previous studies, GR and two GR esters derivatives have been used as additives in a Mater-Bi® type biopolymer, a commercial blend based on TPS developed by Novamont. It has been determined that neat GR provided a plasticising, compatibilising and solubilising effect in the polymeric matrix which facilitates the processability of the material by decreasing the processing temperature. Meanwhile, the GR esters used in previous works improve the miscibility of the TPS-based matrix components and improve its mechanical performance [5,37]. The present work aims to study the effect of GR and some GR derivatives on the TPS matrix obtained from native corn starch. In a first step, native corn starch was mixed with glycerol and water and further melt-extruded at a maximum temperature of 130 °C at high pressure and high shear to obtain TPS. The obtained TPS was then blended either with GR or GR derivatives by melt-extrusion and further injection moulding process. The obtained formulations were mechanically, thermally and structurally characterised to get information of the possible application of these sustainable materials at an industrial level.

## 2. Materials and Methods

### 2.1. Materials

Food grade corn starch, containing 27% amylose was supplied by Cargill (Barcelona, Spain). Distilled water and glycerol (Panreac 99% of purity—Barcelona, Spain) were used as plasticisers. As additives, five pine resins derivatives were used with the following molecular structure (Figure 1):

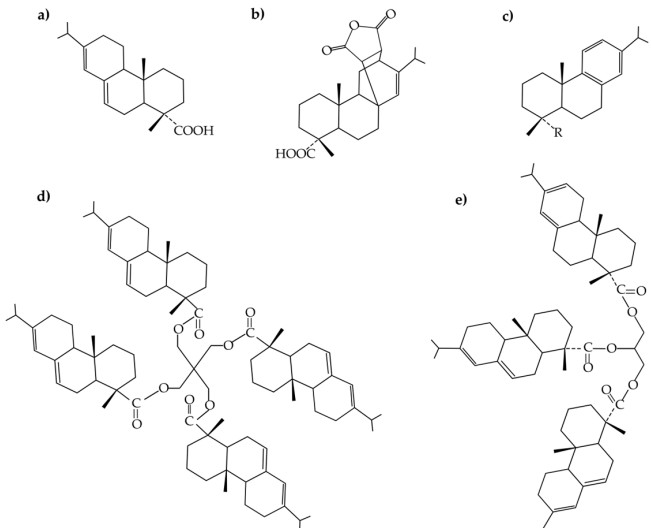

**Figure 1.** Molecular structure of the studied pine resins: (**a**) gum rosin (GR), (**b**) gum rosin modified with maleic anhydride (CM), (**c**) disproportionated gum rosin (dehydroabietic acid) (RD), (**d**) pentaerythritol ester of gum rosin (LF), and (**e**) glycerol ester of gum rosin (UG).

Gum rosin (GR, softening point of 76 °C and acid number 167) was kindly supplied by Sigma-Aldrich (Mostoles, Spain). Gum rosin modified with maleic anhydride under the trade name Colmodif R-330 (CM, softening point of 123 °C and acid number 252), a disproportionated gum rosin (dehydroabietic acid) under the trade name of Residis 455 (RD, softening point of 74.6 °C and acid number 157), a pentaerythritol ester of gum rosin under the trade name of Lurefor 125 (LF, softening point of 125 °C and acid number 11.9) were kindly supplied by LureSA (Segovia, Spain). Glycerol ester of gum rosin under the trade name of Unik Gum G88 (UG, food grade, softening point of 87 °C and acid number 7) was kindly supplied by United Resins (Figueira da Foz, Portugal).

### 2.2. Methods

#### 2.2.1. TPS-Resin Blends Preparation

Figure 2 shows a schematic representation of the preparation and processing of the developed materials and involves: a manual mixing, a first extrusion process to obtain TPS, a second extrusion process to obtain TPS-resin blends and a final injection moulding step to obtain the injected moulded parts.

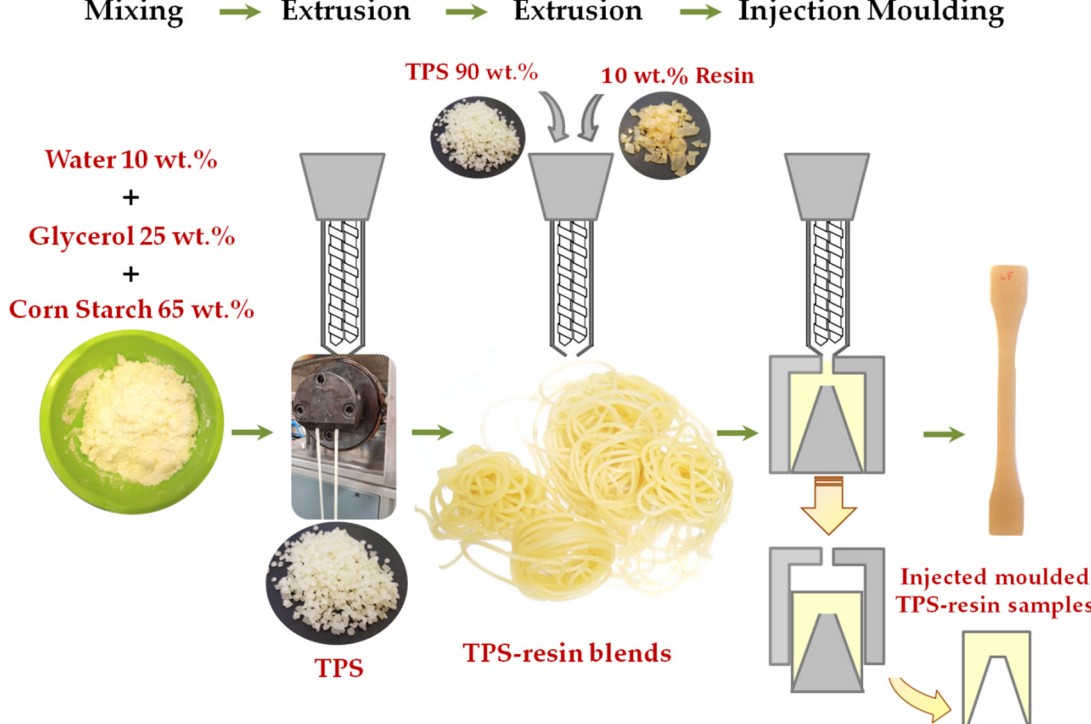

**Figure 2.** Schematic representation of the thermoplastic starch (TPS) preparation and the processing of TPS–resin blend formulations.

In brief, TPS was prepared manually premixing in a hermetically sealed polyethylene bag, 65% of native corn starch with 25 wt. % of glycerol and 10 wt. % of distilled water, 24 h before the extrusion process to ensure homogeneity of the material and to allow the correct diffusion of the plasticisers in the starch matrix [38]. Thus, the corn starch, glycerol and water mixture was processed in a co-rotating twin-screw extruder, L/D ratio of 25 from Dupra S.L (Castalla, Spain) at 20 rpm with a temperature profile of 130, 110, 100, 90, 80 °C (from die to hopper) to obtain TPS. A second extrusion step was performed to prepare the TPS-resin based formulations, adding each pine resin derivatives in 10 wt. % to the previously prepared TPS. Neat TPS was also melt extrusion processed a second time to have the same processing conditions. Thus, six formulations were prepared and named as summarised in Table 1. After the melt extrusion process, the TPS and TPS–resin materials were pelletised and

conditioned at 25 °C and 50 ± 5% of relative humidity (RH) [5]. Then, the sample test specimens were obtained by injection moulding process in an injection moulding machine Sprinter-11, Erinca S.L. (Barcelona, Spain), with a temperature profile of 130, 110, 100 °C, from die to hopper. Prior to characterization, injected moulded samples were conditioned at 25 °C and 50 ± 5% RH [5].

**Table 1.** TPS–resin blend studied formulations and their composition.

| Formulations | Type of Resin in the Blend | Resin Commercial Name | TPS (wt. %) | Resin (wt. %) |
|---|---|---|---|---|
| TPS | - | - | 100 | 0 |
| TPS-GR | Gum rosin or unmodified colophony | Gum rosin | 90 | 10 |
| TPS-CM | Maleic anhydride modified gum rosin | Colmodif R-330 | 90 | 10 |
| TPS-LF | Pentaerythritol ester of gum rosin | Lurefor 125 | 90 | 10 |
| TPS-RD | Disproportionated gum rosin | Residis 455 | 90 | 10 |
| TPS-UG | Glycerol ester of gum rosin | Unik Gum G88 | 90 | 10 |

### 2.2.2. Mechanical Characterization

Tensile test was performed according to standard tests methods ISO 527 [39]. It was carried out in a universal test machine Ibertest ELIB-50-W (Madrid, Spain) with a load cell of 5 kN and a crosshead speed of 100 mm/min. The samples were injected in dog-bone-shaped specimens "1BA" type (80 mm × 10 mm × 4 mm) according to ISO 527 [39]. At least five specimens for each formulation were characterized and the mean and standard deviation of the values are reported. The Young's modulus, the tensile strength and the elongation at break were determined. In addition, to calculate the toughness of the materials, the area under the typical stress-strain curve was calculated. The area was calculated using the OriginPro2015 software from OriginLab (Northampton, MA, USA).

Shore D hardness of all studied formulations was measured on a durometer Model 673-D from Instrument J.Bot, S.A. (Barcelona, Spain) using samples of 4 mm thickness, according to standard tests methods ISO 868 [40]. The mean of 20 measurements at random positions of the samples was reported as the hardness values.

The significant differences in the mechanical parameters were statistically assessed at 95% confidence level according to Tukey's test using a one-way analysis of variance (ANOVA) by means of OriginPro2015 software.

### 2.2.3. Thermal Characterization

Differential scanning calorimetry (DSC) was carried out in a Mettler DSC821e (Toledo, Spain) with a thermal cycles program that consisted in a first heating scan from −50 to 160 °C, followed by a cooling cycle from 160 to −50 °C and a second heating scan from −50 to 200 °C. The heating rate for all cycles was 10 °C/min and the test were performed under a nitrogen atmosphere with a flow of 30 mL/min. The glass transition temperature ($T_g$) was calculated on the second heating scan of the DSC curve.

Thermogravimetric analysis (TGA) was conducted in a TGA PT1000 from Linseis (Selb, Germany). The heating was carried out from 30 to 700 °C at a heating rate of 10 °C/min in a nitrogen atmosphere with a flow rate of 30 mL/min. The degradation onset temperature ($T_{5\%}$) was determined at 5% of mass loss. As well, the degradation endset temperature ($T_{95\%}$) was assessed at 95% of mass loss. The maximum degradation temperature ($T_{max}$) was determined at the peak of the first derivative of the TGA curve (DTG).

### 2.2.4. Structural Characterization

Attenuated total reflectance–Fourier transform infrared spectroscopy (FTIR-ATR) was applied to study the chemical interactions between TPS, GR and gum rosin derivatives (CM, LF, RD and UG). The analysis was performed in a Perkin Elmer Spectrum BX (FTIR system) coupled with an ATR Pike MlRacle ™ (Beaconsfield, United Kingdom). All the formulations as well as neat TPS and the raw resins were assessed within the range of 4000–700 $cm^{-1}$ with a resolution of 16 $cm^{-1}$ and 20 scans.

Scanning electron microscopy (SEM) of the cryo-fractured surface of each formulation was carried out on a Phenon SEM equipment of FEI (Eindhoven, The Netherlands), with a voltage of 5kV. Prior to SEM observation, the samples were coated with a gold-palladium alloy layer to make the sample surface conductive in a Sputter Mod Coater Emitech SC7620, Quorum Technologies (East Sussex, UK).

## 3. Results

TPS was successfully produced from corn starch plasticised with glycerol and water and then it was further used to prepare TPS-resin blends by melt extrusion using GR and GR derivatives. Both neat TPS as well as each TPS-resin compounding were further processed by injection moulding. Injection moulded samples of TPS-resin blend formulations were successfully obtained without any change in the injection moulding processing parameters with respect to neat TPS.

It is known that during the melt-extrusion and injection moulding process, bio-based polymeric matrices can suffer degradation due to the strong shear stresses that act in the viscous molten polymer [22,41]. In fact, injection moulding parameters (i.e., injection speed, barrel temperature and back pressure) significantly affect the impact strength of the polymeric matrices [42]. Thus, injection moulded polymers for rigid packaging are required for high mechanical performance to overcome the strong shear stresses during processing to successfully obtain injected moulded parts as well as to offer the desired mechanical resistance during service. Thus, tensile tests measurements up-to-rupture as well as Shore D hardness were assayed on injected TPS and TPS–resin blend formulations (Figure 3). Results show that TPS is a brittle polymer with a Young's modulus of 11.93 MPa (Figure 3a), a tensile strength of 2 MPa (Figure 3b) and a strain at break of 63.46% (Figure 3c). Young's modulus of TPS-GR, TPS-CM, TPS-RD and TPS-UG have no statistical differences among them nor with TPS (Figure 3a, $p > 0.05$). TPS-LF is the only formulation that presents a statistically higher modulus with respect to TPS ($p < 0.05$), indicating a stiffening effect due to LF addition to the TPS matrix. These results show that LF increased the mechanical properties in the formulation due to a reinforcement effect, which increases the TPS modulus by 55%. This reinforcement effect is because LF increases the cohesion between the starch phases [37]. The ester groups in LF interacted with the starch hydroxyls groups by hydrogen bonding interactions which leads to the compatibilisation of both components [43]. In a previous work we have blended Mater-Bi® type bioplastic (Mater-Bi® NF 866 which is composed of TPS, polybutylene adipate-co-terephthalate (PBAT) and poly($\varepsilon$-caprolactone) (PCL) with LF and we observed that 10 wt. % was not able to significantly improve the Young's modulus of the neat polymeric matrix, while it requires higher amounts (i.e.,: 15 wt. %) to significantly increase it [5]. The higher Young's modulus obtained here shows that LF possess good miscibility with starch matrix, being the only resin able to significantly increase the Young's modulus of TPS. In fact, in a more detailed microscopic analysis it was observed that LF not only possess good miscibility with starch phase of Mater-Bi® NF 866, but also it was able to increase the miscibility between the semi-crystalline and the amorphous phases of thermoplastic starch (TPS) [37]. The TPS tensile strength (Figure 3b) was significantly reduced when all the additives were incorporated to the polymeric matrix, being TPS-LF and TPS-UG the formulations with the highest tensile strength among the studied samples. Regarding the flexibility of the materials, the results show that the elongation at break significant (Figure 3c, $p < 0.05$) increased only when unmodified GR was added to the TPS matrix. GR act as a plasticiser as it increased neat TPS elongation at break in 43%. The plasticising effect of gum rosin has been already observed for other bioplastics such as PLA [44], PCL [45] and Mater-Bi® NF 866 [5]. Meanwhile, the other gum rosin derivatives produce no significant effect on the elongation at break of neat TPS (Figure 3c, $p > 0.05$).

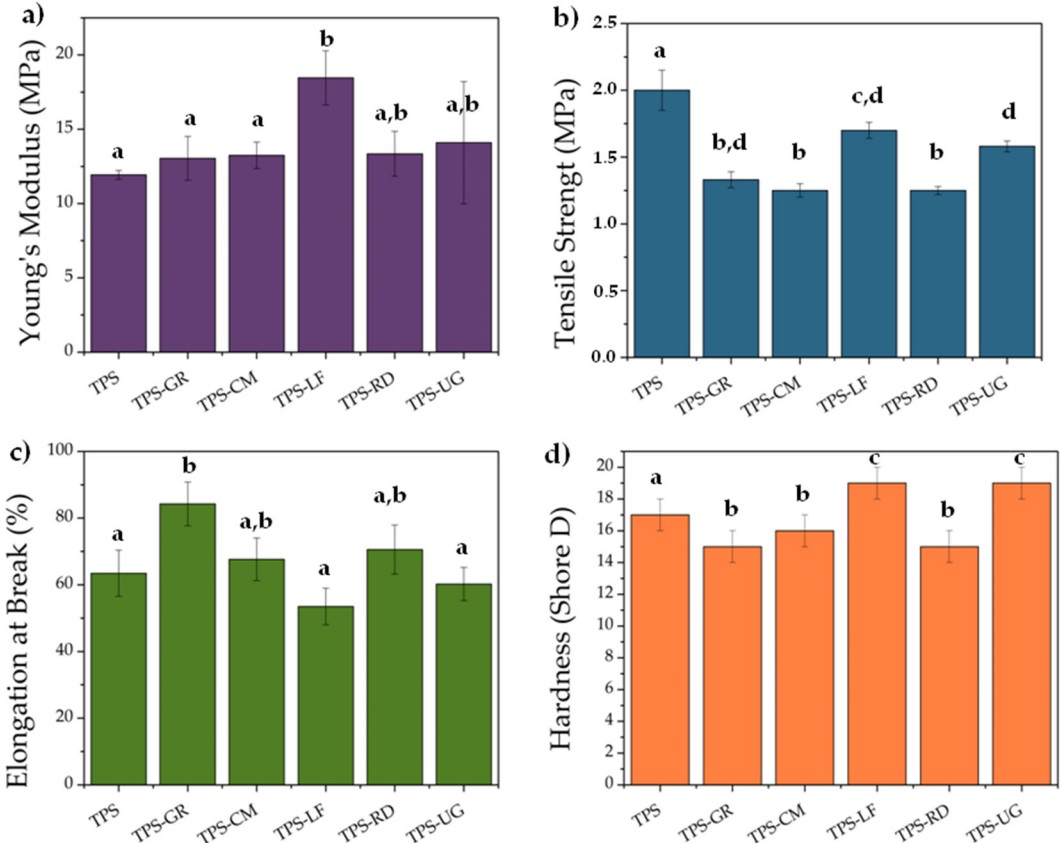

**Figure 3.** Tensile and hardness properties of TPS and TPS-resin formulations: (**a**) Young modulus, (**b**) Tensile Strength, (**c**) elongation at break and (**d**) Shore D. [a–d] Different letters within the same property show statistically significant differences between formulations ($p < 0.05$).

Hardness properties of TPS and the formulations with GR and rosin derivatives were also measured (Figure 3d). All the formulations presented significant differences with TPS Shore D hardness ($p < 0.05$). While the addition of an ester resin (LF or UG) increased TPS hardness values in 12% (from 17 in neat TPS to 19 for TPS-LF and/or TPS-UG), the materials that contain the other resin additives (GR, CM, and RD) reduced TPS hardness values in 12% (from 17 in neat TPS to values up to 15) (Figure 3d). Although the hardness of polymers do not necessarily provide correlate results with other fundamental properties, since it is an empirical quantity related to the inherent indentation resistance [46], it was observed that the TPS-resin blends showed similar tend than the tensile strength values (Figure 3b). Similarly, Ferri et al. (2018) blended PLA with TPS in 70:30 proportion and further added increasing amounts of maleinised linseed oil (from 2 to 8 phr) to increase the compatibility between both biopolymeric matrices, and observed a decrease of Shore D values accompanied by a related decrease on the tensile strength values [28]. Comparing these results with those previously obtained for Mater-Bi® NF 866 blended with 10 wt. % of GR and LF, it was observed that GR decreased the Shore D values of Mater-Bi® NF 866 (around 8%), but it was maintained in Mater-Bi® blended with 10% of LF [5]. These results confirm once again the good miscibility of LF with starch polymeric matrix.

The typical stress–strain curves of the materials with the toughness values of each formulation is shown in Figure 4. It was found that the addition of either GR or rosin derivatives significantly reduced the TPS toughness ($p < 0.05$). The results show that GR has the highest toughness among the studied formulation and that the toughness values present no statistical differences when LF or RD were added ($p > 0.05$). It is important to notice that TPS-LF presents statistically the same elongation at break and a higher Young's modulus than neat TPS (Figure 3a). Therefore, LF provides the highest strength between the studied materials. The same trend was exhibited by UG, since it is an ester of gum rosin. On the other hand, GR statistically maintains TPS Young's modulus and increases its

elongation at break. Thus, TPS-GR would withstand greater tension that neat TPS. The other modified gum rosin additives (CM and RD) exhibit behaviour similar to GR. These results confirm a stiffening and a plasticisation effect produced by LF and GR addition, respectively.

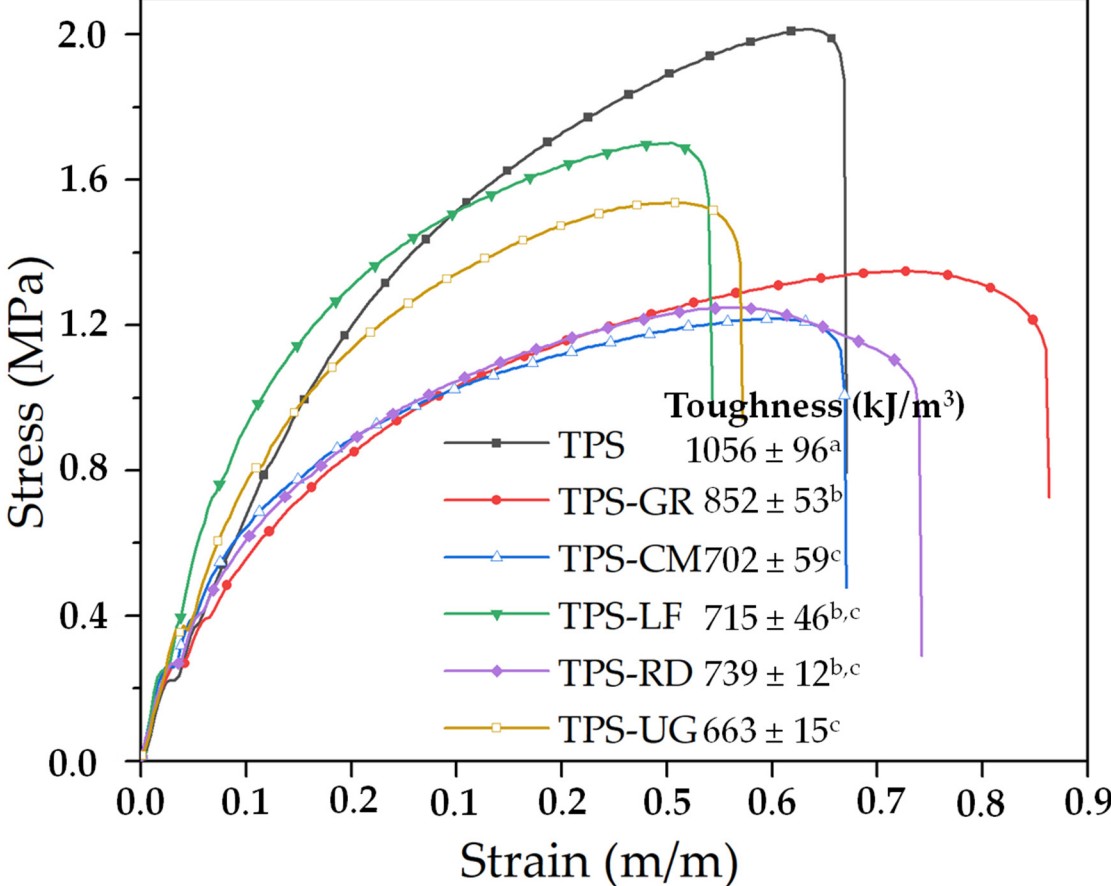

**Figure 4.** Stress–strain curves of TPS and TPS-resin blend formulations with the toughness values. [a–d] Different letters within the same property show statistically significant differences between formulations ($p < 0.05$).

The DSC curves of TPS and its TPS-resin formulations with 10 wt. % of GR and rosin derivatives are shown in Figure 5. $T_g$ of TPS is dependent on glycerol and water contents and it could be found between 34 °C and 70 °C [47,48]. The studied TPS exhibits the $T_g$ at 45.7 °C. However, the addition of gum rosin and rosin derivatives shifts the glass transition temperature to higher temperatures. For TPS-GR and TPS-RD, $T_g$ was found at 56.5 °C and 58.1 °C, respectively. Meanwhile for TPS-LF, TPS-CM and TPS-UG it was found between 70 °C and 80 °C. These results suggest that the formulations present a stiffening effect since GR and its derivatives increase the $T_g$ values, but it was more marked in those formulations blended with UG, CM and LF. The increase on $T_g$ values is very interesting for several applications (i.e., a cup for hot beverages and/or lids for hot food).

GR and RD possess similar chemical structure (see Figure 1). GR is mainly made up of abietane-type acid while RD is made up of pimarane-type acids and is more stable [49]. Therefore, RD side groups are not modified [49], and its chains can move freely between TPS chains lubricating them [5]. On the other hand, CM, LF and UG have chemicals modifications in its chemical side groups (see Figure 1) that hinder its mobility, mainly because they are able to interact with TPS hydroxyl groups [43,49]. However, these chemicals modifications produced a different effect on the mobility of the chains that will further affect the final performance of TPS-resin based formulations. LF is a pentaerythritol ester while UG is a glycerol ester. Thus, TPS-LF would have a higher $T_g$ than TPS-UG, as a pentaerythritol ester has higher cohesion, adhesion, viscosity at fusion and glass transition temperature than a glycerol

ester. On the other hand, CM is a modified gum rosin with maleic anhydride that have reactive sites that differ from those in unmodified gum rosin [49], as CM has three carboxylic acid groups, which increases its acidity [49]. As a result, CM, LF and UG shift $T_g$ to higher temperatures (70 °C or more). The resins with group modifications had a more marked stiffening effect. In fact, the tendency of increment in $T_g$ values follows the same tendency as the Young's modulus. The shifts in TPS glass transition temperatures suggest that GR and its derivatives have good miscibility with TPS matrix [37,50]. With DSC results, it was not possible to establish differences between the $T_g$ of the TPS-GR and TPS-RD. However, it is possible to determine that LF is the modified resin with the highest stiffening effect because it produced the highest displacement of $T_g$. In fact, it shifted the $T_g$ of TPS from 45 °C to 79.3 °C, which is 9 °C more than UG and 7 °C more than CM. The obtained DSC results are consistent with TPS-LF increment of Young's modulus, TPS-LF and TPS-UG increase in hardness, as well as with the plasticisation behaviour of TPS-GR, already discussed.

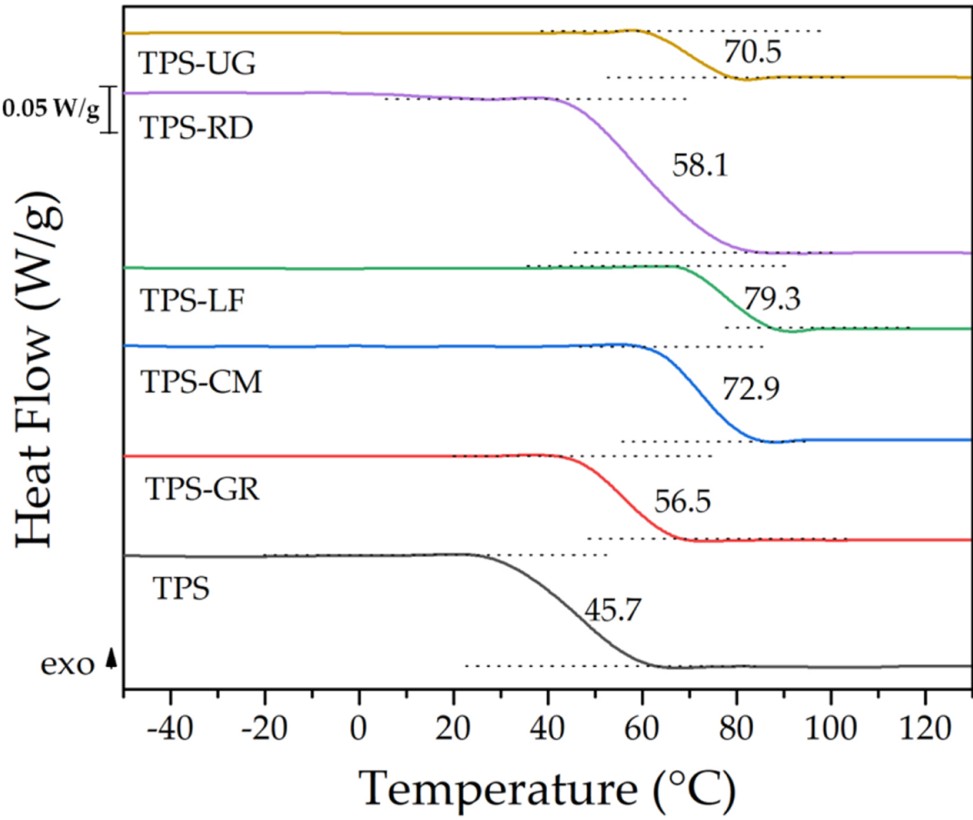

**Figure 5.** Differential scanning calorimetry (DSC) second heating scan curves of neat TPS and TPS-resins blend formulations.

TGA curves and their derivatives (DTG) for TPS and its TPS-resin blend formulations are shown in Figure 6. The thermal values at different degradations stages, determined by TGA analysis, are shown in Table 2. It can be noticed that the formulated samples were more thermally stable than the neat TPS in all the degradation stages. TGA curves showed a three-step degradation process (Figure 6a). Between 100 °C and 170 °C an initial weight loss of 12% occurs, which is related with the bound water content in the TPS samples [51,52]. The degradation onset temperature ($T_{5\%}$) of TPS (Table 2) increased with the addition of GR and rosin derivatives. GR rises $T_{5\%}$ by 29 °C, CM by 23 °C, LF does by 25 °C, while RD and UG produce the lowest increase in $T_{5\%}$ by 16 °C and 11 °C, respectively. Therefore, the chemical structures of the additives allow them to positively interact with the TPS matrix, providing a thermal stabilising effect [5]. Nevertheless, GR and LF create a higher protection to thermal degradation than the other additives. The increased thermal stability has been related to the

positive interaction between the carbonyl groups of the pine resins and the hydroxyl groups of the starch material (i.e., hydrogen bonding interactions) [53].

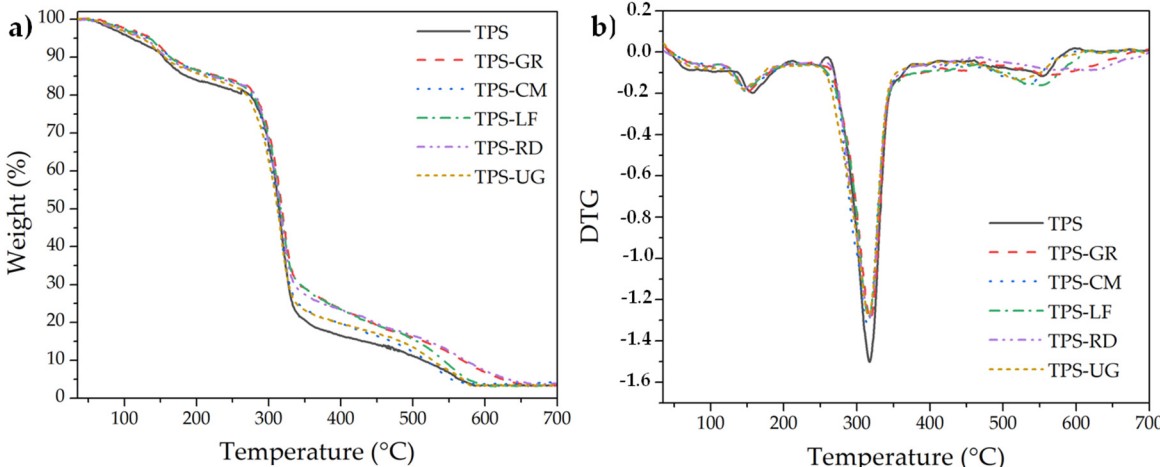

**Figure 6.** (**a**) TGA and (**b**) their derivatives (DTG) curves of neat TPS and TPS-resin blend formulations.

**Table 2.** TGA thermal parameters degradation onset temperature ($T_{5\%}$), temperature of maximum degradation rate ($T_{max}$) and degradation endset temperature ($T_{95\%}$) for TPS and the TPS-resin blend formulations.

| Material | $T_{5\%}$ (°C) | $T_{max}$ (°C) | $T_{95\%}$ (°C) |
|----------|----------------|----------------|-----------------|
| TPS | 109.3 ± 2.1 | 317.3 ± 1.9 | 564.8 ± 2.3 |
| TPS-GR | 138.3 ± 1.9 | 318.3 ± 1.7 | 624.3 ± 2.1 |
| TPS-CM | 132.8 ± 2.0 | 315.8 ± 1.9 | 553.8 ± 1.9 |
| TPS-LF | 133.8 ± 1.9 | 316.3 ± 1.7 | 576.3 ± 2.3 |
| TPS-RD | 126.3 ± 1.8 | 316.8 ± 2.0 | 632.2 ± 1.8 |
| TPS-UG | 121.8 ± 2.0 | 315.8 ± 1.8 | 570.2 ± 2.0 |

The second step of degradation starts at 260 °C and it is the main degradation process. In this step the degradation is produced because of the depolymerization of starch [51], that is the cleavage of ether linkages in starch backbone and it is also related with the degradation of the starch/glycerol reach phase [52]. During this degradation step the dehydration of neighbouring hydroxyl groups in the glucose ring take place, resulting in the formation of C-C bonds or breakdown of the glucose ring with the further formation of aldehyde end groups [27].

The DTG curve (Figure 6b) shows that the maximum degradation temperature ($T_{max}$) did not present differences between all the materials with a mean value around 316 °C (316.6 ± 1.8 °C). Even so, the effect in the mass loss was appreciable in the formulations that contain gum rosin and derivatives. This effect can be seen in Figure 6a, at 350 °C. The TPS reports a mass loss of 20%. In the case of TPS blended with GR, LF and RD reduction in the mass loss of 29%, 29% and 27% were observed, respectively. The incorporation of CM and UG to TPS showed the mass loss of TPS in 23%. GR, LF and RD created interactions and links with TPS matrix, because of the presence of carbonyl and ester groups, which protect the polymeric matrix from thermal degradation [5]. Besides, CM resin produced less effect in the thermal protection because the spatial distribution of its molecules avoids the creation of a lot of links between resin and TPS polymeric matrix. It is important to notice that even when UG resin has carbonyl groups on its structure, the effect in the thermal stability is lower that the LF resin which possesses one more carbonyl group and higher molecular weight. At 335 °C the final degradation step takes place, where organic residues decompose and turn into ashes. It is seen that all the pine resins protected TPS matrix from thermal degradation, with GR, LF and RD providing the greatest stability in the degradation endset temperature.

The FTIR spectra comparison of neat TPS and TPS-resin blends formulations with 10 wt. % of gum rosin (GR) and rosin derivatives (CM, LF, RD, UG) is presented in Figure 7.

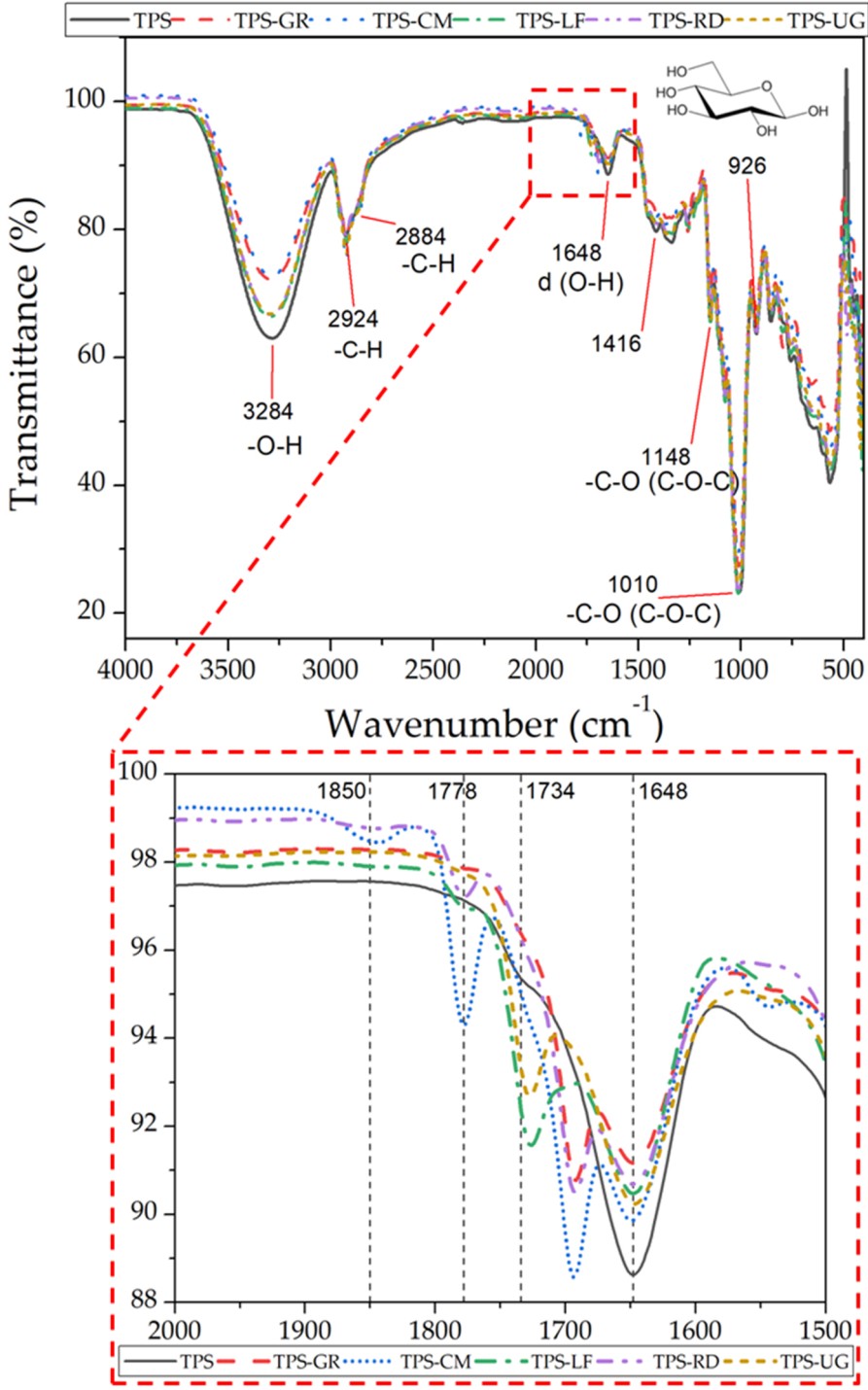

**Figure 7.** Fourier transform infrared spectroscopy (FTIR) spectra with representative peaks of net TPS and its comparison with TPS-resin blend formulations. A zoom in the range between 2000 cm$^{-1}$ and 1500 cm$^{-1}$ is also shown.

The FTIR spectrum of neat TPS presents the typical absorption bands corresponding to the functional groups of starch and glycerol. The band corresponding to C-O stretching of C-O-C bond was found at 1010 cm$^{-1}$ and 1148 cm$^{-1}$ and the C-O stretching of pyranose rings at 926 cm$^{-1}$. The bound water band was located at 1648 cm$^{-1}$ (δ (O-H)). The O-H groups are found at 3284 cm$^{-1}$ and C-H

stretching is located at 2924 cm$^{-1}$ [54–57]. The representative bands of the glycerol plasticizer were displayed at 2884 cm$^{-1}$(C-H), associated with the hydroxyl groups, as well as at 1416 cm$^{-1}$ and at 1148 cm$^{-1}$ (C-O stretching), associated with carbon-oxygen (C–O) absorption peaks characteristic of primary and secondary alcohols [19,54,56].

It is seen that the spectra of the formulations are similar to that of neat TPS. This is expected since the resins content is 10 wt. %. Dang and Yoksan (2015) have reported a similar behaviour when working with starch films blended with chitosan added in 0.37–1.45 wt. % [57]. Further, there are no significant shifts in the characteristic peaks of TPS, which indicates that the interaction with the resins is not strong enough to be detectable with this technique in accordance with Mendes et al. (2017) [54]. Nevertheless, two main differences can be observed in the spectra of TPS-resin blend formulations with respect to neat TPS. The first one is referred to the reduction of the -OH group band related with the bound water, found at 3284 cm$^{-1}$ and at 1648 cm$^{-1}$. A reduction in this peak indicates that when TPS is blended with resins, the -OH groups of TPS have more affinity with resins than with water. Therefore, -OH groups are involved in positives interaction with GR and/or rosin derivatives such as hydrogen bonding [57]. The second difference was found in the region between 2000 cm$^{-1}$ and 1500 cm$^{-1}$ (see zoom in Figure 7). In TPS-CM signals of -C=O stretch of maleic anhydride appeared as low intensity peaks at 1780 cm$^{-1}$ and 1850 cm$^{-1}$ [58]. In TPS-LF and TPS-UG a low intensity band was found at 1727 cm$^{-1}$. This band is representative of the -C=O stretching of the ester group [43]. In TPS-GR and TPS-RD the peak of stretching of the carboxylic group (-C=O) is found at 1692 cm$^{-1}$ [5]. Those displacement of carbonyl group band suggest an hydrogen bonding interactions between TPS and the respective resins [5,59].

Finally, the morphology and microstructure of the cryo-fractured surfaces of TPS as well as the effect the gum rosin presence on the TPS matrix were studied by SEM (Figure 8). No apparent phase separation is observed in the SEM image of the cryo-fractured surfaces of plasticized corn starch with glycerol and water (Figure 8a), confirming the homogeneous dispersion of glycerol and water in corn starch matrix and the successful formation of TPS. Moreover, neat TPS shows a smooth fracture surface typical of TPS with amylose content around 25–30% [52]. No apparent phase separation was observed in TPS-resin blends regardless the resin used, indicating that all resins are well incorporated to the TPS matrix (Figure 8b–f). All TPS-resin blend formulations showed a much rougher behaviour in accordance with the increased stiffening effect. Among al TPS-resin blend formulations, TPS-GR (Figure 8b) and TPS-RD (Figure 8e) show much cracked surface. Moreover, in TPS-GR (Figure 8b), some micro-voids are observed (see inset Figure 8b), which are probably responsible of the reduction of the mechanical resistance of this formulation as reveals the marked reduction of the tensile strength (Figure 3b) and Shore D hardness (Figure 3d). Similarly, TPS-RD (Figure 7e) shows some micro-domains that in some cases shows phase debonding (see inset Figure 8e), with the corresponding reduction of the tensile strength (Figure 3b) as well as Shore D hardness (Figure 3d). Meanwhile, in TPS-CM (Figure 8c), TPS-LF (Figure 8d) and TPS-UG (Figure 8f) formulations, resin particles appear homogeneously dispersed as small spherical domains (see inset Figure 8c for TPS-CM, inset Figure 8d for TPS-LF and inset Figure 8f for TPS-UG). In a previous work, Mater-Bi$^{®}$ NF 866 was blended with 10 wt. % of LF and SEM micrographs shows poor interfacial adhesion between the polymeric phases since small domains with empty interface were observed [5]. Thus, the results obtained here suggest that LF possess good miscibility with starchy matrix (see inset Figure 8d). In fact, in a more detailed microscopic analysis somewhat improvement in the miscibility between the amorphous and semi-crystalline phases of thermoplastic starch due to the LF gum rosin derivate addition was observed for the already commented Mater-Bi$^{®}$ NF 866 blended with 10 wt. % of LF [37].

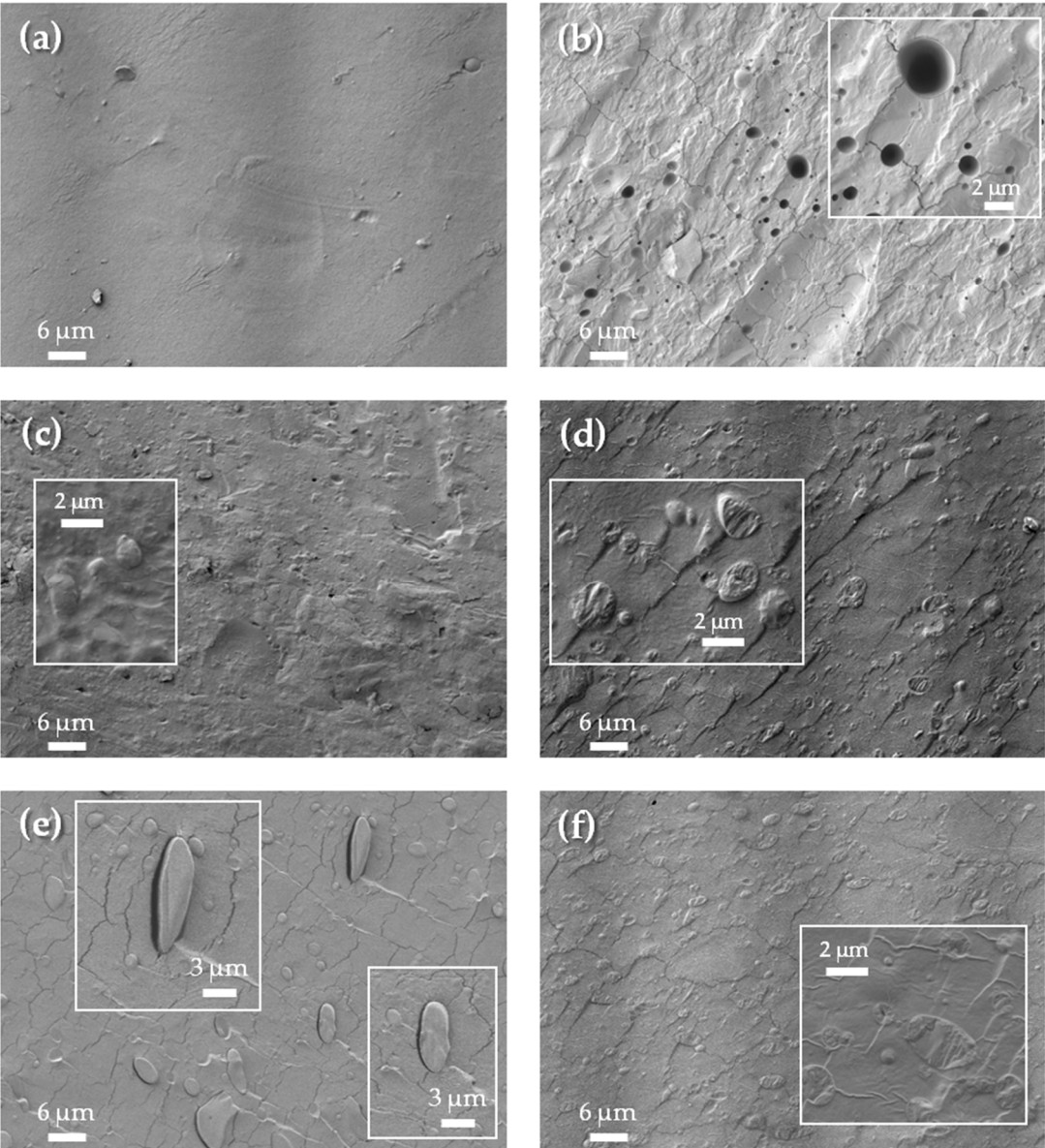

**Figure 8.** Scanning electron microscopy (SEM) images of the crio-fractured surface of: (**a**) TPS, (**b**) TPS-GR, (**c**) TPS-CM, (**d**) TPS-LF, (**e**) TPS-RD and (**f**) TPS-UG.

## 4. Conclusions

Homogeneous TPS based materials were prepared starting with corn starch and using water as well as glycerol as plasticisers. The obtained TPS was then blended with pine resin derivatives by melt extrusion and further by injection moulding process. All TPS-resin blend formulations showed higher thermal stability than TPS counterpart, improving the processing window of TPS. FTIR spectra showed that gum rosin and its derivatives established hydrogen bond interactions with the hydroxyl groups of starch and glycerol in TPS matrix. Interactions were stronger in the case of blends with higher amounts of carbonyl groups in their chemical structure such as CM, UG and particularly LF. In fact, although all resins produce a stiffening effect, CM, UG and LF generated a significant stiffening effect, shifting the $T_g$ of TPS from 45 °C to higher temperatures greater than 70 °C. Consequently, in TPS-CM, TPS-UG and TPS-LF the Young's modulus slightly increase while were among the TPS-resin based formulations with the highest hardness values following the same trend as the $T_g$ increment.

Therefore, it can be concluded that gum rosin and particularly modified gum rosin derivatives showed good compatibility with the developed TPS from corn starch and allowed to develop

homogeneous fully bio-based materials, ensuring thermal stability for melt-extrusion and injection moulding processing as well as a significant stiffening effect. Consequently, these bio-based materials show their potential for industrial applications such as disposable cups for hot beverage and/or lids containers for hot food applications.

**Author Contributions:** Conceptualization, M.A. and M.P.A.; methodology, M.A.; validation, M.A. and C.P.; formal analysis, M.A. and C.P.; investigation, M.A., C.P., M.P.A. and J.L.-M.; resources, J.L.-M.; data curation, M.A., C.P. and M.P.A.; writing—original draft preparation, M.A., C.P. and M.P.A.; writing—review and editing, M.A. and M.P.A.; visualization, M.A., C.P., M.P.A. and J.L.-M.; supervision, J.L.-M.; project administration, M.P.A. and J.L.-M. funding acquisition, M.P.A. and J.L.-M. All authors have read and agreed to the published version of the manuscript.

**Funding:** This research was funded by the Spanish Ministry of Economy and Competitiveness (MINECO), project: PROMADEPCOL (MAT2017-84909-C2-2-R) as well as by Santander-UCM (PR87/19-22628) project. M.A. thanks Secretaria Nacional de Educación Superior, Ciencia, Tecnología e Innovación (SENESCYT-Ecuador) and Escuela Politécnica Nacional. C.P. thanks Santiago Grisolía fellowship (GRISOLIAP/2019/113) from Generalitat Valenciana and M.P.A. thanks MINECO for her postdoctoral contract: Juan de la Cierva-Incorporación (FJCI-2017-33536).

**Conflicts of Interest:** The authors declare no conflict of interest.

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
