# Peer review of "Pine Resin Derivatives as Sustainable Additives to Improve the Mechanical and Thermal Properties of Injected Moulded Thermoplastic Starch"

_applsci, doi:10.3390/app10072561_

Round 1

Reviewer 1 Report

General Comments

The originality of the article is appreciated. However, the choice of words and phrases is a challenge to understanding the authors’ views and arguments. To overcome these challenges suggestions have been made to the authors. The work resulted in a composite material but the title does not indicate this. Figure 7 needs to be marked up/annotated so that the features discussed by authors under results is clear to the reader. It is suggested that authors use images with higher magnifications so that these features are easier to identify.

Specific Comments

LINE 2-4: The work resulted in a composite material.

Hence, I suggest that the title be modified to indicate this reality thus: "Pine resin derivatives as sustainable additives to improve the mechanical and thermal properties of injection moulded thermoplastic starch composite”.

LINE 26:  Insert "of" between "are" and "interest"

LINE 38:  Replace "focus" with "focused"

LINE 39:  Replace "lids containers " with "container lids"

LINE 43:  Replace "not suitable" with "unsuitable "

LINE 52: Replace "Being corn the" with "Corn is the"

LINE 55-56:  Rephrase sentence and get rid of the phrase "but also"

LINE 56:  Replace "has not a" with "does not exhibit"

LINE 57:  Replace "to introduce it" with "to use starch"

LINE 61:  Replace "there are required" with "requires"

LINE 71-72: Rephrase sentence that starts with “Moreover,…..” for clarity.

LINE 72-73:  Replace "TPS industrial applications still has limitations because of" with "TPS industrial application pf TPS is limited by"

LINE 79:  Delete "In this sense"

LINE 79:  Insert "a" between "is" and "natural"

LINE 96: The meaning of the words “processing performance” is not clear.  Do you mean the ease of processing?? If the answer is yes, replace "processing performance of" with "ease of processing"

LINE 96-97:  Instead of "the used of GR esters improve the miscibility among the components of the matrix " write "the used of GR esters improves the miscibility of the matrix components"

LINE 99:  Instead of "coming" use "obtained".

LINE 140:  Replace "extruded” with "extrusion"

LINE 142:  Delete "it is"

LINE 143:  What is the meaning of "HR"? It is relative humidity the correct abbreviation is "RH" and the should be written both in full and in abbreviation the first time it appears in the text.

LINE 146:  Instead of "Previously to the" write "Prior to"

LINE 154:  Instead of "rate" use "speed"

LINE 156:  Instead of "break" use "fracture"

LINE 165:  Insert "software" after "OriginPro 2015"

LINE 175:  Replace "onset degradation temperature" with "degradation onset temperature"

LINE 176:  Replace "endset degradation temperature " with "degradation endset temperature"

LINE 177:  The phrase "maximum degradation rate temperature" is not familiar. Please confirm from previous works and rephrase accordingly.

LINE 184:  Both the resolution and the number of scans used are quite low?? It will be good to justify the use of these parameters.

LINE 188:  Replace "Previously " with "Prior to SEM observation"

LINE 195:  Use "injection" instead of "Injected"

LINE 197-198: Rephrase the sentence for better clarity. The phrase "high mechanical performance " is very vague in the context.

LINE 199: Delete "to"

LINE 206:  Use "increases" instead of "increment "

LINE 211: Use "significantly" instead of "significant "

LINE 212:  Use "significantly" instead of "significant "

LINE 214:  Use "more detailed" instead of "deeper”

LINE 217:  Use "significantly" instead of "significant "

LINE 224:  Use "fracture" instead of "break"

LINE 227-228:  he sentence "While the addition of a esters resin (LF or UG) increased TPS hardness in 12 %, 227 the rest of the resin additives (GR, CM, and RD) reduced TPS hardness in 12 %. " lacks clarity. especially the use of "in 12%". Do you mean 12% increase/decrease or at 12% While the addition of a esters resin (LF or UG) increased TPS hardness in 12 %, 227 the rest of the resin additives (GR, CM, and RD) reduced TPS hardness in 12 %. concentration?

LINE 231:  Use "once" instead of "ones"

LINE 241:  Insert "values" between "toughness" and "present".

LINE 243:  Use "fracture" instead of "break".

LINE 244:  Use "trend" instead of "tendency "

LINE 246:  Use "fracture" instead of "break".

LINE 247:  Instead of "similar behaviour than " write "behaviour similat to "

LINE 255:   Use "on" instead of "of"

LINE 255:  Do you mean the water content in the formulation? If yes, Instead of "humidity content", the use of "water content" might be more appropriate.

LINE 268:  Rephrase the phrase "as it is know that GR does", as it does not fit well for scientific literature.

LINE 269:  Instead of "difficult" use "affects" or "impedes"

LINE 270-271:  The phrase "these modifications have 270 different degrees" lacks clarity. Please rephrase

LINE 277:  Use "the tendency of increment in" instead of "the tendency of the increment in"

LINE 278:  Use "as" instead of "than"

LINE 279-280:  Rewriite the sentence "DSC did not allow to establish differences between GR and RD " thus " With DSC results it was not possible to d establish differences between GR and RD"

If you mean differences in Tg of GR and RD, please indicate this in the sentence.

LINE 286:  Use "for" instead of "of".

LINE 293:  Write "bound " instead of "bounded "

 LINE 294:  Write "degradation onset temperature " instead of "onset degradation temperature "

LINE 297:  Write "allow" instead of "allows "

LINE 299:  Write "to" instead of "with"

LINE 315-316:  Write "reduction in the mass loss of 29 %, 29 % and 27% were observed " instead of "it was observed a reduction in the mass loss in 29 %, 29 % and 27% "

LINE 324:  Delete "they"

LINE 325:  Instead of "being GR, LF and RD the ones that provide the greatest stability " write "with GR, LF and RD providing the greatest stability"

LINE 331:  Write "bound " instead of "bounded "

LINE 347:  Write "bound " instead of "bounded "

LINE 371:  Features discussed in Fig. 7 are generally not easy to distinguish. Use higher magnification images and annotate the images to show the features mentioned in the discussion like micro-voids in Figure 7b.

LINE 379: Use "more detailed" instead of "deeper".

LINE 389:  Instead of "Those interactions were strongly" Write "Interactions were stronger"

LINE 392:  Instead of "than 70 °C.”,  write "greater than" 70oC"

LINE 394:  Instead of "tendency of ", write "trend as"

LINE 397:  Instead of "biobased materials "write "bio-based composite materials"

LINE 398:  Instead of "biobased materials "write "bio-based composite materials"

Author Response

Reviewer’s Comments for Article titled:  Pine resin derivatives as sustainable additives to improve the mechanical and thermal properties of injected moulded thermoplastic starch

General Comments

The originality of the article is appreciated. However, the choice of words and phrases is a challenge to understanding the authors’ views and arguments. To overcome these challenges suggestions have been made to the authors. The work resulted in a composite material but the title does not indicate this. Figure 7 needs to be marked up/annotated so that the features discussed by authors under results is clear to the reader. It is suggested that authors use images with higher magnifications so that these features are easier to identify.

We thank reviewer for his/her valuable comments as well as for appreciate the article originality. We have taken into account the considerations performed by the reviewer an each of them has been reply. Nevertheless, one point should be clarified. Our materials are blend formulations instead of composites materials. Although the purpose of both, blends and composite materials, is to obtain a product able to combine the positive properties of each constituent component, blends formulations are developed by physical blending two or more polymeric materials, while composites consist in a polymeric matrix reinforced with fillers with micronized dimensions. These statements have now clarified in the current version of the manuscript. We think that now it is clearer improving the overall quality of the manuscript. Figure 7 has been updated introducing magnifications of specific areas of the fracture surface of the materials.

Specific Comments

LINE 2-4: The work resulted in a composite material.

Hence, I suggest that the title be modified to indicate this reality thus: "Pine resin derivatives as sustainable additives to improve the mechanical and thermal properties of injection moulded thermoplastic starch composite”.

As it was already clarified the developed formulations are blends instead of composites.

LINE 26:  Insert "of" between "are" and "interest"

LINE 38:  Replace "focus" with "focused"

LINE 39:  Replace "lids containers " with "container lids"

LINE 43:  Replace "not suitable" with "unsuitable "

LINE 52: Replace "Being corn the" with "Corn is the"

LINE 55-56:  Rephrase sentence and get rid of the phrase "but also"

LINE 56:  Replace "has not a" with "does not exhibit"

LINE 57:  Replace "to introduce it" with "to use starch"

LINE 61:  Replace "there are required" with "requires"

Thank you for all these observations. All these sentences have been now corrected.

LINE 71-72: Rephrase sentence that starts with “Moreover,…..” for clarity.

The sentence has been now extended to better explain the concept of circular economy, which in turn is the topic of the Special Issue.

LINE 72-73:  Replace "TPS industrial applications still has limitations because of" with "TPS industrial application pf TPS is limited by"

LINE 79:  Delete "In this sense"

LINE 79:  Insert "a" between "is" and "natural"

Thank you for all these observations. All these sentences have been now corrected.

LINE 96: The meaning of the words “processing performance” is not clear.  Do you mean the ease of processing?? If the answer is yes, replace "processing performance of" with "ease of processing"

These sentences have been changed to clarify this point.

LINE 96-97:  Instead of "the used of GR esters improve the miscibility among the components of the matrix " write "the used of GR esters improves the miscibility of the matrix components"

It refers to the GR used, not to the use of GR esters. Thus, the GR esters used improve the miscibility of the matrix components and improve its mechanical performance

LINE 99:  Instead of "coming" use "obtained".

LINE 140:  Replace "extruded” with "extrusion"

LINE 142:  Delete "it is"

LINE 143:  What is the meaning of "HR"? It is relative humidity the correct abbreviation is "RH" and the should be written both in full and in abbreviation the first time it appears in the text.

LINE 146:  Instead of "Previously to the" write "Prior to"

LINE 154:  Instead of "rate" use "speed"

Thank you for all these observations. All these sentences have been now corrected.

LINE 156:  Instead of "break" use "fracture"

We thank this suggestion. However, the mechanical property determined in this work has been the elongation at break, which is a terminology more frequently used and appropriate in the field of application.

LINE 165:  Insert "software" after "OriginPro 2015"

It has been inserted in the current version of the manuscript.

LINE 175:  Replace "onset degradation temperature" with "degradation onset temperature"

It has been replaced.

LINE 176:  Replace "endset degradation temperature " with "degradation endset temperature"

It has been replaced.

LINE 177:  The phrase "maximum degradation rate temperature" is not familiar. Please confirm from previous works and rephrase accordingly.

It has been replaced by maximum degradation temperature, which is more frequently used in this field.

LINE 184:  Both the resolution and the number of scans used are quite low?? It will be good to justify the use of these parameters.

We always use these resolution parameters for FTIR analysis, and we obtain enough resolution with our FTIR spectrometer.

LINE 188:  Replace "Previously " with "Prior to SEM observation"

LINE 195:  Use "injection" instead of "Injected"

Thank you for all these observations. All these sentences have been now corrected.

LINE 197-198: Rephrase the sentence for better clarity. The phrase "high mechanical performance " is very vague in the context.

We thank reviewer this comment. We have now extended these general statement to introduce the lecturer to the importance of the processing parameters on the mechanical response of the injected moulded materials.

LINE 199: Delete "to"

LINE 206:  Use "increases" instead of "increment "

LINE 211: Use "significantly" instead of "significant "

LINE 212:  Use "significantly" instead of "significant "

LINE 214:  Use "more detailed" instead of "deeper”

LINE 217:  Use "significantly" instead of "significant "

Thank you for all these observations. All these sentences have been now corrected.

LINE 224:  Use "fracture" instead of "break"

As it was already commented we have measured the elongation at break.

LINE 227-228:  he sentence "While the addition of a esters resin (LF or UG) increased TPS hardness in 12 %, 227 the rest of the resin additives (GR, CM, and RD) reduced TPS hardness in 12 %. " lacks clarity. especially the use of "in 12%". Do you mean 12% increase/decrease or at 12% While the addition of a esters resin (LF or UG) increased TPS hardness in 12 %, 227 the rest of the resin additives (GR, CM, and RD) reduced TPS hardness in 12 %. concentration?

This sentence has now clarified.

LINE 231:  Use "once" instead of "ones"

LINE 241:  Insert "values" between "toughness" and "present".

LINE 243:  Use "fracture" instead of "break".

LINE 244:  Use "trend" instead of "tendency "

LINE 246:  Use "fracture" instead of "break".

LINE 247:  Instead of "similar behaviour than " write "behaviour similat to "

LINE 255:   Use "on" instead of "of"

Thank you for all these observations. All these sentences have been now corrected.

LINE 255:  Do you mean the water content in the formulation? If yes, Instead of "humidity content", the use of "water content" might be more appropriate.

This statement has been changed for the proposed one.

LINE 268:  Rephrase the phrase "as it is know that GR does", as it does not fit well for scientific literature.

This part of the phrase has been deleted.

LINE 269:  Instead of "difficult" use "affects" or "impedes"

Thank you for your suggestion. It has been changed by hinder which is more appropriate in this context.

LINE 270-271:  The phrase "these modifications have 270 different degrees" lacks clarity. Please rephrase

This phrase has been changed, extended and improved

LINE 277:  Use "the tendency of increment in" instead of "the tendency of the increment in"

LINE 278:  Use "as" instead of "than"

Thank you for these observations. Both sentences have been now corrected.

LINE 279-280:  Rewriite the sentence "DSC did not allow to establish differences between GR and RD " thus " With DSC results it was not possible to d establish differences between GR and RD"

If you mean differences in Tg of GR and RD, please indicate this in the sentence.

Thank you for this observation. It has been now indicated

LINE 286:  Use "for" instead of "of".

LINE 293:  Write "bound " instead of "bounded "

 LINE 294:  Write "degradation onset temperature " instead of "onset degradation temperature "

LINE 297:  Write "allow" instead of "allows "

LINE 299:  Write "to" instead of "with"

LINE 315-316:  Write "reduction in the mass loss of 29 %, 29 % and 27% were observed " instead of "it was observed a reduction in the mass loss in 29 %, 29 % and 27% "

LINE 324:  Delete "they"

LINE 325:  Instead of "being GR, LF and RD the ones that provide the greatest stability " write "with GR, LF and RD providing the greatest stability"

LINE 331:  Write "bound " instead of "bounded "

LINE 347:  Write "bound " instead of "bounded "

Thank you for all these observations. All these sentences have been now corrected.

LINE 371:  Features discussed in Fig. 7 are generally not easy to distinguish. Use higher magnification images and annotate the images to show the features mentioned in the discussion like micro-voids in Figure 7b.

We thank Reviewer for his/her comment. We introduce magnifications as inset figures and we think that now the figure is clearer, improving the overall performance of the discussion of SEM results.

LINE 379: Use "more detailed" instead of "deeper".

LINE 389:  Instead of "Those interactions were strongly" Write "Interactions were stronger"

LINE 392:  Instead of "than 70 °C.”,  write "greater than" 70oC"

LINE 394:  Instead of "tendency of ", write "trend as"

Thank you for all these observations. All these sentences have been now corrected.

LINE 397:  Instead of "biobased materials "write "bio-based composite materials"

LINE 398:  Instead of "biobased materials "write "bio-based composite materials"

Although both terminologies, bio-based and biobased, are frequently and interchangeably used, we have changed biobased by bio-based as reviewer suggested. Nevertheless, as it was already commented we have developed blends materials instead of composites.

Reviewer 2 Report

This draft manuscript is complete with an overall satisfactory explanation and discussion of the data. Some minor points have to be addressed before it is accepted for pubblication. Please see attached file.

Author Response

Points to be addressed:

Abstract Line 22: please define better thermal stability

We thank Reviewer for this suggestion. In the current version of the manuscript we have extend this statement.

Abstract Line 23: please define better the interaction between carbonyl group and hydroxyl groups

We thank Reviewer for this suggestion. In the current version of the manuscript these interactions (hydrogen bond interactions) were better explained.

Pag 2 Line 61: change “there are” with “these features are”

Pag 5 Line 154: Please specify all the sizes of the specimens and if they were ASTM standards.

Pag 6 Line 206: Please change “increment” with “increases”

Pag 9 Line 295: change “in” with “by”

We thank Reviewer for all these observations. All these sentences have been now corrected.

Pag 9 Line 301: please define better the positive interaction between carbonyl group and hydroxyl groups

We thank Reviewer for this suggestion. In the current version of the manuscript, the positive interactions have been defined as hydrogen bonding interactions.

Table 2: please explain why TPS-UG having the lowest Young’s modulus has also the highest hardness (1st column vs 4th column)

We thank reviewer for this observation. We have extended the discussion of the mechanical results. Actually, TPS-UG did not have the lowest Young Modulus, since it is statically equal to that of TPS. However, this sample posses a high standard deviation and it seems that it is lower. Therefore, in the current version of the manuscript, we have adjusted the results using five dog-bone shaped samples and we have now expressed all the mechanic results (Young Modulus, Tensile Strength and Elongation at break as well as Shore D hardness) in a Figure 3 with the corresponding standard deviation and ANOVA analysis to better visualize the obtained results. Moreover, we have now related the mechanical results with the microstructural analysis performed by SEM. We think that the results are now more evident and clearer and, thus, improving the overall performance of the manuscript.
